# Single Molecule Fluorescence Spectroscopy of PSI Trimers from *Arthrospira platensis*: A Computational Approach

**DOI:** 10.3390/molecules24040822

**Published:** 2019-02-25

**Authors:** Roman Pishchalnikov, Vladimir Shubin, Andrei Razjivin

**Affiliations:** 1Prokhorov General Physics Institute of the Russian Academy of Sciences, 119991 Moscow, Russia; 2Bach Institute of Biochemistry of the Russian Academy of Sciences, 119071 Moscow, Russia; shubin@inbi.ras.ru; 3Belozersky Research Institute of Physico-Chemical Biology, Moscow State University, 119992 Moscow, Russia; razjivin@gmail.com

**Keywords:** photosynthesis, PSI, long-wavelength antenna chlorophyll, energy transfer, single molecule spectroscopy, modified redfield theory, excitons

## Abstract

Based on single molecule spectroscopy analysis and our preliminary theoretical studies, the linear and fluorescence spectra of the PSI trimer from *Arthrospira platensis* with different realizations of the static disorder were modeled at cryogenic temperature. Considering the previously calculated spectral density of chlorophyll, an exciton model for the PSI monomer and trimer including the red antenna states was developed taking into account the supposed similarity of PSI antenna structures from *Thermosynechococcus e.*, *Synechocystis sp. PCC6803*, and *Arthrospira platensis*. The red Chls in the PSI monomer were assumed to be in the nearest proximity of the reaction center. The PSI trimer model allowed the simulation of experimentally measured zero phonon line distribution of the red states considering a weak electron-phonon coupling for the antenna exciton states. However, the broad absorption and fluorescence spectra of an individual emitter at 760 nm were calculated by adjusting the Huang-Rhys factors of the chlorophyll lower phonon modes assuming strong electron-phonon coupling.

## 1. Introduction

The pigment-protein complexes (PPCs) of photosynthetic species are perfect objects for consideration if one wants to demonstrate how the mutual arrangement of cofactors in the system can affect its optical properties. Generally, the PPCs of higher plants, algae, and bacteria are localized in thylakoid membranes and consist of a protein skeleton supporting either chlorophylls (Chl) or bacteriochlorophylls, carotenoids, and some inorganic molecules that ensure the proper functioning of the photosynthetic machinery [1]. Most PPCs can be divided into one of two types: the reaction centers (RCs) and the light-harvesting complexes (LHC). The LHCs include up to tens of pigments per protein and absorb the Sun’s radiation in the visible and infra-red ranges, transforming it into excited states called excitons. The absorbed energy in the form of excitons eventually transfers to RCs, where the charge separation takes place [2,3,4,5]. The RCs usually consists of six cofactors, and in some cases they can be isolated from surrounding LHCs.

Photosystem I (PSI) of plants and cyanobacteria supports light-induced electron transfer from the lumen to the stromal side of the membrane. Along with photosystem II (PSII), cytochrome b6f, and ATP synthase, PSI is an indispensable component of the photosynthetic apparatus. Converting solar energy into chemical bond energy, PSI and PSII act in tandem. The amount of PSI in cyanobacteria is relatively higher in comparison with PSII. The PSI/PSII ratio varies from about 3 up to 5.8 [3,6]. PSII absorbs a photon with wavelength less than 680 nm, creating the cation radical P680^+^, which drives the sequential oxidation of water. PSI in turn captures light energy up to 700 nm. The oxidizing potential for the primary electron donor of PSI is not enough to obtain electrons from water; however, it is enough to support electron transfer between cytochrome and ferredoxin. Thus, along with the linear electron transport from water to NADP^+^, PSI supports cyclic electron transport through the membrane [7], which plays an important role in photoprotection [8,9].

Unlike the cyanobacterial PSI, the plant PSI consists of a core complex and four subunits that are called the outer antenna LHCI. PSI of cyanobacteria has no outer antenna and exists in both monomeric and trimeric forms [3]; however, some tetrameric forms of PSI have been found recently [10]. There are two bacteria for which a high-resolution PSI trimer crystal structure exists. These are *Thermosynechococcus elongatus* [11] and *Synechocystis sp. PCC6803* [12] at 2.5 Å resolution. These data revealed remarkable similarity of the PSI structure of the different species. Analysis of a few differences of these structures allows speculating about the possible arrangement of PSI trimers and monomers of others bacteria, for instance, *Arthrospira platensis* [13,14,15], whose optical properties have already been studied thoroughly.

When compared with other PPCs, PSI has characteristic absorption bands in the long-wavelength part of its spectrum, which is strange because they are lower in energy than the primary electron donor P700 of the RC [14,15,16]. In this case, according to the Boltzmann distribution, energy trapping does not appear to be the most probable channel for the energy flow. There are several explanations for these low energy or red antenna states. It is generally supposed that they likely appear due to Chl site energy variations and strong exciton coupling in some groups of Chls in PSI [11,17,18,19], called long-wavelength Chls (LWCs). Such PSI red states belong not only to monomers, but to trimers too. The reddest states so far have been found in the PSI trimer from *Arthrospira platensis* [20,21]. The lowest absorption band of the PSI trimer peaks around 740 nm and has fluorescence at 740 nm (F740) and 760 nm (F760) at cryogenic temperatures. It must be stressed that the 740 nm band disappeared in absorption of monomeric PSI, which can be explained by the emergence of strong exciton interaction between the side Chls of monomers after trimerization. Surprisingly, the fluorescence intensity of the LWCs of PSI trimer from *Arthrospira platensis* depends on the redox state of P700. The F760 peak is much weaker than F740 when the P700 of the RC is oxidized [22,23].

Studying the relationship between optical properties of LHCs and their structure requires the use of many spectroscopic techniques [24]. Single molecule fluorescence spectroscopy (SMS) stands out from the many experimental methods not only by the way of sample preparation, but also of the experimental data interpretation [25,26,27]. A sample under study is usually in a buffer solution, but in the case of SMS, it has to be immobilized on a surface. The isolated PPCs exhibit fluorescence blinking, which is influenced by slight changes of protein conformation and variations of coupling with the local environment [28]. The main feature of SMS is the possibility to observe the fluorescence signal from a single LHC and better signal-to-noise ratio in comparison with linear absorption measurements [28]. The standard measurements give an optical response averaged over a large number of complexes, and the contributions of a particular molecular unit are hidden behind the response of an ensemble. The fluctuations of the band peaks in the emission spectra are a direct visualization of the site energy changes due to interaction with a protein skeleton. Depending on the time resolution of an SMS registration system, different physical phenomena such as protein dynamics (up to seconds), radical pair recombination (ms-μs), singlet-singlet and singlet-triplet annihilation (ns-ps), and energy transfer (fs) can be investigated [28].

SMS measurements at cryogenic temperatures give more information about the pure physical aspects: the LHC exciton manifold properties and exciton-phonon interaction, while measurements at physiological temperatures allow observing an optical response that is much closer to the natural biological processes in the LHC [29,30,31,32]. Concerning studies of LWCs, SMS applications have provided insight into the polarization of the red emitters [33,34]. To simulate the SMS signal from a PSI complex of any species and at any temperature, a detailed exciton model of the energy transfer and charge separation has to be developed. Fitting of the linear and the steady-state fluorescence spectra of PSI from *Thermosynechococcus elongatus* based on Redfield theory [17] has already been done. One of the advantages of this research is that the site energies of Chls have been determined by using the genetic optimization algorithm. There is a study that used modified Redfield theory to model energy transfer and trapping kinetics in PSI [35]. In both cases, only the monomeric exciton Hamiltonian (96 pigments) was taken into account. Some thorough calculations on PSI, when the site energies were calculated applying various modifications of semiclassical approximations, have also been performed [18,19,36,37,38,39]. In these studies, several solutions for the two main difficulties of making the theory of PSI have been proposed: Chl Qy site energy identification, assessment of LWCs, and Chl mutual interaction energy calculation. The main drawback of almost all these studies is a poor elaboration of the exciton theory, which is necessary to model the realistic optical response. Without additional consideration and analysis of the electron-phonon coupling and the relaxation rates of the PSI exciton manifold, it will not be possible to link together the numerous experimental data and the theoretical estimates of Qy transition energies of Chls and the couplings between them.

The aim of this research is to propose a reasonable interpretation of the LWC fluorescence dynamics at cryogenic temperatures of the PSI trimer isolated from *Arthrospira platensis*, taking into account the existing crystal structure for *Thermosynechococcus* and *Synechocystis* as well as our preliminary studies on PSI exciton models [40]. Considering the outcomes of the SMS experimental work on PSI form *Arthrospira platensis* [14], we implement in our theory the different regimes of electron-phonon coupling for the excitons and for the LWCs. To achieve the desired goal for the first time, we will apply the modified Redfield theory for the PSI trimer exciton model development and the multimode Brownian oscillator model for the Q_y_ absorption of Chl modeling.

## 2. PSI Structure

Because no crystal structure of PSI from *Arthrospira platensis* has been reported, we overview the basic elements of PSI based on the available data. According to the PSI structure from *Thermosynechococcus* [11], the PSI monomer consists of 12 protein subunits PsaA, PsaB, PsaC, PsaD, PsaE, PsaF, PsaI, PsaJ, PsaK, PsaL, PsaM, and PsaX. The total number of pigment molecules is 118: 96 Chls and 22 carotenoids. Also, there are two phylloquinones, four lipids, and three Fe_4_S_4_ clusters. All of the cofactors are covalently bound to the 12 protein subunits. Most of the pigments of the PSI core complex, including cofactors of the electron transport chain, are located in the large PsaA and PsaB subunits (Figure 1). All Chls are coordinated by histidines.

Each of two subunits contains 11 transmembrane helices whose special arrangement is quite symmetric with respect to each other [41]. The PsaC, PsaD, and PsaE proteins are located on the stromal side of PSI. They secure the proper docking site for the electron acceptor. The PsaC subunit holds two of the three iron-sulphur clusters, which are coordinated by cysteines. The PsaD and PsaE subunits are directly involved in stabilization of PsaC. PsaF, PsaJ, PsaK, and PsaX are located at the distal site of PSI. PsaF has a rather unusual protein structure and, most probably, together with PsaJ, mediates the energy transfer from the outer antenna. The PsaJ subunit in turn coordinates three Chls and stabilizes PsaF. PsaX most probably belongs only to thermophilic cyanobacteria. It coordinates one Chl and is in hydrophobic contact with several carotenoids.

When grown under low light conditions, cyanobacterial PSI forms a trimeric complex to increase the efficiency of light absorption [41]. The PSI trimer has a molecular weight about 1 MDa. The key structural elements of a core complex providing the PSI trimerization are the PsaL, PsaI, and PsaM protein subunits. PsaL and PsaI form a so-called trimerization domain in the center of a trimer, while a PsaM subunit occupies the space between monomers. There are two features connected with PsaL and trimerization: a Ca^2+^ ion is coordinated by PsaA and PsaL amino acids on the luminal side (Figure 1); and the PsaL loops are in close contact with the PsaD subunit on the stromal side [41]. PsaM is a small subunit that coordinates only one chlorophyll molecule and is in contact with one carotenoid.

## 3. Comparison of Different PSIs

To model the optical properties of a PSI isolated from a cyanobacterium with unknown crystal structure, the similarities and differences of already known cyanobacterial PSI structures must be discussed. The PSI trimer from *Synechocystis sp. PCC6803* has been recently crystalized [12]. It was found that the newly crystalized structure is in some ways different from that of the well-known *Thermosynechococcus* [11]. First of all, the total number of Chls in the core complexes differs by one: 95 in *Thermosynechococcus* and 96 in *Synechocystis*. There are five major differences in Chl locations and protein surrounding that might be crucial for the fluorescence SMS simulations (Figure 2). M1601 Chl from the PsaM subunit is missing in *Synechocystis*; it is substituted by a lipid from a neighboring protein (Figure 2b). Also, the whole PsaX subunit is absent in *Synechocystis* (Figure 2f). The pigment from PsaX is replaced by F203 Chl from PsaF. It is curious that the Chl trimers in *Thermosynechococcus*, which are commonly recognized as red chlorophylls, became dimers in *Synechocystis* (Figure 2d,e). However, cofactors of the electron transport chain and its local environment remain untouched in the two species.

## 4. *Arthrospira platensis* PSI Trimer

The fact that PSI from *Arthrospira platensis* exists in thylakoid membranes in monomeric and trimeric forms was reported many years ago [21]. The electron microscopy technique allowed the estimation of the PSI geometry. It appeared that the PSI oligomeric complex is a disk-shape particle 19–20 nm in diameter divided into three sectors. This means that the size of the PSI trimer from *Arthrospira platensis* is practically identical to that of *Thermosynechococcus* and *Synechocystis*. Taking into account the similarity of cyanobacterial PSI crystal structures discussed in the previous chapter and the findings of these studies [21], we suggest that the PSI monomer crystal structure is much the same as those of *Thermosynechococcus* or *Synechocystis* and can differ only by a few pigment molecules and small protein subunits. Thus, for the modeling of optical properties of the PSI monomer isolated from *Arthrospira platensis*, we adopt the Chl locations from 1JB0.pdb, however allowing ourselves to add or remove some molecules of the structure to get a better fitting of the experimental data.

## 5. Results

### 5.1. Chl Monomer

Before the PSI modeling, the Chl absorption and fluorescence in the protein environment must be simulated. The evaluation of the g(t) function allows us to calculate the temperature dependent spectra. In the limit T→0 K, Equation (3) turns into the zero phonon line equation [42]. A proper interpretation of the cryogenic SMS fluorescence spectra of PSI without knowledge of the Chl zero phonon line function is impossible. The 4 K Chl absorption and the spectral density (Figure 3) was calculated using 49 Brownian oscillator modes $\left\{S_{j},\omega_{j},\gamma_{j}\left(\omega \right)\right\}$ listed in [43] and in Table A1. To integrate Equations (1) and (3) over *t* at 4 K, $2^{14}=16384$ points of the time scale were applied. However, to model the optical properties of any PPC, it is not necessary to keep such time array since the inhomogeneous broadening and the exciton-exciton relaxation allow acceptable accuracy of integration with much fewer points.

The distribution of the maximum in the emission spectrum is one of the characteristic results in SMS. Parameters of the distribution are described as the effect of the nearest environment on the electronic states of a pigment. On the example of monomeric Chl in solution, we will illustrate how the resulting fluorescence depends on the observation time. The zero phonon line and the calculated inhomogeneous broadened spectra are shown on the same plots in Figure 4. Instead of using the analytical expression for the linear spectra, we have employed the Monte Carlo procedure and have simulated $\Omega_{eg}$ modulations caused by the environment.

### 5.2. Linear Spectra of PSI

The calculated linear spectra of PSI monomers and trimers are shown in Figure 5. The sizes of the exciton Hamiltonian for monomer and trimer models are 96 × 96 and 288 × 288, respectively. The inhomogeneous broadening is incorporated into the computational procedures using the Monte Carlo method. The FWHM of inhomogeneous broadening for the exciton absorption and fluorescence spectra is 180 cm^−1^. For each state the corresponding $g_{\alpha \beta \mu \nu }\left(t\right)$ line-shape function, the $\lambda_{\alpha \beta \mu \nu }$ reorganization energies, and the $K_{\mu \nu }$ relaxation matrix are estimated using the data from Table A1 and Table A2. Taking into account the conclusions in reference [14], the electron-phonon coupling for the exciton states is considered to be weak. To satisfy this condition we decreased the initial $S_{1}$ and $S_{2}$ Huang-Rhys factors from Table A1 to the values of 0.32 and 0.04, respectively.

The crucial point of the calculations is the value of the temperature used for evaluation of Equations (13), (14), and (15). At the extremely low temperatures, about 1–3 K, almost all integrals diverge. This obstacle can be overcome by increasing the size of the integration timescale. However, the computational costs are increased much faster than would be expected. For this study, keeping in mind 96 exciton states for the monomer and 288 for the trimer, we found that only at 30 K the final results can be obtained for a reasonable time with the available computing resources. In comparison with the calculations for monomeric Chl in solution, the PSI modeling was done using a timescale of $2^{10}=1024$ points. Thus, we did not try to get a perfect fit of the absorption and CD data by applying the multiparametric optimization procedure since the temperatures of the measured data and the calculated ones do not coincide. In fact, it does not produce a significant impact on the final conclusions (especially for fluorescence), since the experiments, done at cryogenic temperatures in the range of several dozen degrees, are not very different with each other.

### 5.3. Fluorescence Spectra of PSI Trimer

Based on the monomer model, the exciton steady-state fluorescence spectra of trimeric complexes were calculated. Results with different number of static disorder realizations are presented in Figure 6. Averaging over 1500 realizations gives a practically smooth symmetric lineshape, and the contribution of the higher states is not so pronounced. Only 50 realizations still allow us to see the effect from the higher states in the spectrum. The distribution of the lowest exciton states and states of the main absorption band at 680 nm are also presented (Figure 6).

Following the results discussed in reference [14], we calculated the absorption and fluorescence of the reddest state measured in SMS experiments at 760 nm. Analysis of these data revealed two features that characterize the broad emission at 760 nm or the F760 state. First, it was shown that the coupling between states of the main exciton manifold and F760 is weak and the energy transfer between them is very slow; second, in comparison with the exciton states, the electron-phonon coupling in F760 is incomparably strong. Considering these findings, we modified the Chl spectral density to get the effect of strong interaction with the bath and about 20 nm Stokes shift for F760. The simulated spectra and zero phonon line are shown in Figure 7. The lowest Huang-Rhys factors (Table A1) were set to $S_{1}=1.5$, $S_{2}=0.4$, $S_{3}=0.3$, and $S_{1}=0.2$; the rest remained the same. The static disorder was set to 200 cm^−1^. By fitting the F760 emission band, we obtained $\Omega_{eg}=13950$ cm^−1^ (about 717 nm). The total PSI trimer absorption and fluorescence spectra were calculated as a sum of the excitonic part and the F760 contribution. Setting the ratio between intensities of the excitonic part and F760 as $I_{F760}^{abs}/I_{ext}^{abs}=0.03$ for absorption and $I_{F760}^{fl}/I_{ext}^{fl}=0.5$ for fluorescence, we got the final spectra that are presented in Figure 8.

## 6. Discussion

The absence of the PSI crystal structure for *Arthrospira platensis* leaves much room for speculations concerning energy transport in the antenna and the charge transfer dynamics. The specific feature of PSI from any species is the existence of actively absorbing states below P700 and the broad fluorescence emitters. Examining the available crystal structures (Figure 2), we realized that the differences in spatial arrangement of the pigments of different cyanobacterial PSIs can be very small. Interestingly, these differences are not in the mutual orientation of pigments but the absence of some Chl, or, maybe it is in the case of *Arthrospira platensis*, the presence of additional molecules compared with the well-known structure of PSI from *Thermosynechococcus*.

Simulation of the linear spectra is a necessary procedure, since standalone SMS fluorescence modeling is not enough to get a proper set of system parameters. The calculated linear absorption and CD spectra are shown in Figure 5. To spare computational time, we did not use the multiparametric optimization procedure like those we have already applied [43]. As a starting point, we used the set of site energies obtained in [17]. This study focused on the simultaneous fitting of the spectra and succeeded in the development of an exciton model based on the Redfield approximation. Primarily, the borrowed site energies were blue shifted by about 400 cm^−1^, which roughly corresponded to the reorganization energy of Chl, then several $\Omega_{n}$ were adjusted by hand to fit the experimental data. Basically, for Chls that are not considered as the red ones, the site energy variability is in the range of 100–150 cm^−1^ (Table A2). The overall fitting of absorption lineshape is satisfactory; however, the characteristic shape of the main absorption band peak was fitted poorly. Actually, this was expected since the fitting was done without optimization. The fitness of CD spectra looks a bit worse than those of absorption. The calculated zero point of the PSI monomer and trimer CD coincides with the experimental one. The CD spectrum of PSI is nonconservative, and to overcome this problem additional terms can be considered in the equation for the rotational strength [17]. We did not use this method; thus, the positive part of the calculated CD spectra of monomer and trimer is more intensive in our modeling than those measured. Moreover, the monomer positive part is spilt, which is not good, but in the trimer CD such split disappeared. We consider that the CD fitting problems are also connected with a lack of optimization of the system parameters and can be improved.

The location of red Chls and their precise role in energy transfer and charge separation are still under debate. We have proposed several conjectures on the phenomenon of LWCs in PSI monomer and trimer. A thorough examination of the Chl arrangement in PSI monomer allows us to conclude that there are some traits in the orientation of Chls relative to the membrane plane. First of all, most of the Chls are located closer to the stromal and luminal sides of the PSI complex; let us call them the outer Chls. The number of so-called inner Chls is rather small; if cofactors of RCs are not taken into account, approximately ten pigment molecules are situated at the same distance from stroma and lumen (Figure 9). Moreover, the inner Chls are located exclusively in close proximity of RCs. Intriguingly, the Qy transition moment of all inner Chls are practically perpendicular to the membrane plane, while the outer Chls and, what is important, the RC cofactors lie in the membrane plane. So, we assume that the putative red Chls could be located around the RC and our model of PSI monomer and trimer form *Arthrospira platensis* does not include the trimeric B31-B32-B33 Chls, as is accepted for *Thermosynechococcus*.

In most studies devoted to cyanobacterial PSI modeling, it is accepted that the interaction between antenna Chls plays an important role in identification of red Chls. Considering this point, various Chl dimers and trimers located in the PSI monomeric complex are regarded as putative candidates as red Chls because of strong excitonic coupling between them. Meanwhile, it has to be stressed that in these works the linear spectra are calculated by dressing the transition moment intensities with Gaussians [18,19,38]. It is clear that such simplification of the simulation of optical properties negates all efforts spent on the assessment of $\Omega_{n}$ site energies either from the first principle or using semiclassical approximations. Based on our theoretical findings, we claim that the interaction energy between PSI antenna pigments cannot be a reason the makes antenna Chls red. The $J_{mn}$ values in our modeling were calculated applying the extended dipole approximation. The maximum and the minimum of $J_{mn}$ for PSI trimer are 196 cm^−1^ and −231.12 cm^−1^. Taking into account that FWHM of the inhomogeneous broadening is 180 cm^−1^ (in different studies it varies in the range of 200–250 cm^−1^), it is impossible to get a significant shift of the spectral bands. Our previous theoretical work on PSI trimers [40], where the effect of trimerization was studied, showed that variations of the interaction energies between Chls of monomers in trimer do not produce the desired influence on fluorescence bands.

After numerous trials we realized that the only way to calculate reasonable PSI linear and fluorescence spectra is to reduce by about 400 cm^−1^ the site energies of some Chls, which are considered to be the red ones. It has to be noted that such shift is practically two times more than the maximum of the exciton coupling. The possible red Chls were chosen from the immediate environment of the RC. As discussed before, this assumption arises from analysis of our experiments on removing of the peripheral Chls form PSI monomers and trimers, and also from the fact that the far-red fluorescence intensities depend on the redox state of P700. Interestingly, the CD spectra were very sensitive to the choice of red Chls. For example, B22, B34, B35, B36, A24, A35, A36, and A37 were tested as red Chls in different combinations; however, none of them gave the proper CD spectrum. Finally, we found that A40, B39, B26, A28, and L01, L02 are most suitable for the current set of experimental data.

Moreover, we can argue that the red exciton states, defined by the adjustment of Qy site energies, are responsible only for the fluorescence signal in the range of the ZPL distribution. The analysis of the SMS studies allows us to expect, most likely, some additional Chls in the *Arthrospira platensis* PSI structure, a few more than 96. By fitting the contribution of F760 to the total fluorescence spectrum, we have determined the transition energy for this state, which can be considered as the site energy of a virtual Chl molecule (Figure 7).

The width of PSI trimer exciton fluorescence, however, is not satisfactory if we compare with the averaged SMS spectra [14]. There is a study in which the interaction of exciton states with the charge transfer states was proposed to describe the abnormal broadening of the red emitters [39]. Our reasonable explanation of this fact is that the F726 band is a complex one and consists of two contributions: the weak electron-phonon excitonic states and the strong electron-phonon red state. It was shown that the F726 and F760 intensity distributions are not correlated [14], and the transition moments of these emitters are perpendicular to each other. All these facts show that red emitters have very low coupling with the exciton manifold. According our modeling, A40, B39, B26, and A28 and the L01 and L02 molecules are responsible only for the excitonic contribution and partially contribute to the total lineshape of F726.

Thus, considering the expanded dipole approximation for the interaction between Chls and setting particular pigment molecules as red Chls, we have reproduced the exciton fluorescence spectrum of PSI trimers that coincides with the ZPL distribution. The experimentally measured ZPL distribution in the fluorescence spectra showed that the emitting exciton states with weak electron-phonon coupling are localized in the range from 695 to 740 nm. To simulate a proper exciton dynamics, we employed the modified Redfield theory with the realistic Chl spectral density. This theory gives comprehensive equations for the exciton relaxation matrix and allows calculating the Stokes shift for each exciton state explicitly.

## 7. Computational Procedures

### 7.1. Chl Absorption Modeling

To simulate a realistic PSI trimer optical response, the Chl linear optical properties must be modeled with maximum accuracy. For the purpose of single molecule spectroscopy, it is enough to consider only the Qy transition of Chl. The absorption and steady state fluorescence spectra of a two-level (ground and excited states) monomeric pigment based on the cumulant expansion method [44,45] are written as follows:(1)$$\sigma_{abs}=\frac{1}{\pi }Re\overset{\infty }{\underset{0}{\int }}dte^{i\left(\omega -\Omega_{eg}\right)t}e^{-g\left(t\right)}e^{-\frac{1}{2}{\left(SD_{\Omega }t\right)}^{2}}$$
(2)$$\sigma_{fl}=\frac{1}{\pi }Re\overset{\infty }{\underset{0}{\int }}dte^{i\left(\omega -\Omega_{eg}+2\lambda \right)t}e^{-g^{*}\left(t\right)}e^{-\frac{1}{2}{\left(SD_{\Omega }t\right)}^{2}}$$
where $\Omega_{eg}$ is the transition frequency, $SD_{\Omega }=FWHM_{\Omega }/2\sqrt{2ln2}$ is the standard deviation of the Gaussian distribution that must be taken into account to incorporate the inhomogeneous broadening with $FWHM_{\Omega }$, λ is the reorganization energy. The temperature dependent line-shape function is g(t):(3)$$g\left(t\right)=\frac{1}{2\pi }\overset{+\infty }{\underset{-\infty }{\int }}d\omega \frac{1-\cos\omega t}{\omega^{2}}\coth\left(\frac{\beta \hbar \omega }{2}\right)C\left(\omega \right)-\frac{i}{2\pi }\overset{+\infty }{\underset{-\infty }{\int }}d\omega \frac{\sin\left(\omega t\right)}{\omega^{2}}C\left(\omega \right)$$
where β=1/kT, T is the temperature, and k is the Boltzmann constant.

The line-shape function includes the C(ω) nuclear spectral density of the $\Omega_{eg}$ electronic transition. The most effective way to estimate the influence of nuclear motions to an electronic transition is to apply the multimode Brownian oscillator model [24]. In this case, the spectral density is given by:(4)$$C\left(\omega \right)=\underset{j}{\sum }\frac{S_{j}\omega \gamma_{j}\left(\omega \right)}{{\left(\omega_{j}^{2}-\omega^{2}\right)}^{2}+\omega^{2}\gamma_{j}^{2}\left(\omega \right)}$$

The parameters of the vibronic Hamiltonian for a particular mode are implicitly collected in three characteristic values: $S_{j}$ is the Huang-Rhys factor, $\omega_{j}$ is the energy of a mode, and $\gamma_{j}\left(\omega \right)$ is the spectral distribution of the electron-phonon coupling or a damping factor. In our modeling of the PSI optical properties, we will use the set of parameters $\left\{\Omega_{eg},FWHM_{\Omega },S_{j},\omega_{j},\gamma_{j}\left(\omega \right)\right\}$ that we have already determined for the Chl a molecule in dimethyl ether [43]. $\left\{S_{j},\omega_{j},\gamma_{j}\left(\omega \right)\right\}$ are listed in Table A1.

### 7.2. Theory of Optical Response for PSI

We consider a system of N interacting two-level pigment molecules to simulate the monomeric and trimeric PSI linear optical response, particularly a steady-state fluorescence signal, which is the basis of the single molecule spectroscopy interpretation. Also, we will assume that only the Qy transition of Chl contributes to the observed signal. Among several theories of different complexity, we will employ the modified Redfield theory, which incorporates the site energy static disorder and the strong exciton-phonon interaction [24,46,47]. According to this theory, for each exciton state the g(t) line-shape function and the relaxation rates between states of the exciton manifold have to be evaluated. Depending on the spectral density, these calculations can be time consuming, and the use of computer clusters becomes a necessary condition for PSI modeling.

The system Hamiltonian includes the exciton $H_{ext}$, phonon $H_{ph}$, and exciton-phonon $H_{ext-ph}$ parts:(5)$$H_{sys}=H_{ext}+H_{ph}+H_{ext-ph}$$

The exciton Hamiltonian is written in terms of the exciton creation $B_{n}^{+}$ and annihilation $B_{n}$ operators for the *n*th Chl molecule, which obey the following commutation rules: $\left[B_{m},B_{n}^{+}\right]=\delta_{nm}\left(1-2B_{m}^{+}B_{m}\right)$:(6)$$H_{ext}=\hbar \underset{n}{\sum }\Omega_{n}B_{n}^{+}B_{n}+\frac{\hbar }{2}\underset{n\neq m}{\sum }J_{mn}\left(B_{m}^{+}B_{n}+B_{n}^{+}B_{m}\right)$$
where $\Omega_{n}$ is the Qy transition energy of the *n*th Chl, n=1…N, and $J_{mn}$ is the interaction between the *n*th and *m*th Chl. $J_{mn}$ can be calculated invoking the dipole-dipole or the extended dipole approximations. The latter is preferable since it gives the correct coupling values at distances less than 10 Å [19,40]. The local interaction of a particular Chl with the bath, which represents a protein environment and molecular phonons, is described by the multimode Brownian oscillator model [24]. The exciton-phonon part of the system Hamiltonian is given by:(7)$$H_{ext-ph}=\underset{nm}{\sum }q_{mn}\left(t\right)B_{m}^{+}B_{n}$$
where $q_{mn}\left(t\right)$ are the collective coordinates, whose averaged values and time evolution are calculated from the $H_{ph}$ free phonon Hamiltonian [24]. The off-diagonal exciton-phonon coupling is considered as a perturbation, while the diagonal part is treated non-perturbatively. The off-diagonal terms depend on the exciton wave function overlapping and the intensity of electron-phonon coupling. Such conditions allow us to get realistic exciton rates in comparison with the Foerster and Redfield regimes [48]. The Fourier transformation of the autocorrelation function of $q_{mn}\left(t\right)$ gives the spectral density in the site representation:(8)$$C_{mnkl}\left(\omega \right)=\frac{i}{2}\overset{\infty }{\underset{-\infty }{\int }}dte^{i\omega t}\left[q_{mn}\left(t\right),q_{kl}\left(0\right)\right]$$

The semiclassical spectral density $C_{mnkl}\left(\omega \right)$ is calculated by using Equation (4). Next, the line-shape function $g_{mnkl}\left(t\right)$ is evaluated numerically according to (3). It is well-known that the interaction of a complex quantum system with laser fields produces an optical response that depends on an intricate net of coupling forces between the components of a system. The linear optical properties of PSI are entirely determined by the single exciton {α} manifold. The system Hamiltonian (6) can be recast in the exciton representation after solving the eigenstate problem. Let $\epsilon_{\alpha }$ and $c_{n}^{\alpha }$ be the eigenstates and eigenvectors of (6), then $B_{\alpha }^{+}|0\rangle =\sum_{n}c_{n}^{\alpha }B_{n}^{+}|0\rangle$ and $B_{\alpha }|0\rangle =\sum_{n}c_{n}^{\alpha }B_{\alpha }|0\rangle$. Greek letters α, β, μ, and ν run from 1 to N indexing the exciton states. The transition dipole moment in the exciton representation is $d_{\alpha }=\sum_{n}c_{n}^{\alpha }d_{n}$, where $d_{n}$ is a vector of the Qy transition of a monomeric Chl. The spectral density (8) and the line-shape function (3) transform into $C_{\mu \nu \alpha \beta }\left(\omega \right)=\sum_{mnkl}c_{m}^{\mu }c_{n}^{\nu }c_{k}^{\alpha }c_{l}^{\beta }C_{mnkl}\left(\omega \right)$ and $g_{\mu \nu \alpha \beta }\left(t\right)=\sum_{mnkl}c_{m}^{\mu }c_{n}^{\nu }c_{k}^{\alpha }c_{l}^{\beta }g_{mnkl}\left(t\right)$.

To calculate the optical response function according to modified Redfield theory, the doorway-window representation is used [24]. It is assumed that after interaction with the laser pulse, an exciton population is created, and then it evolves in time according to the set of master equations [46]:(9)$$\frac{\partial }{\partial t}G_{\mu \nu }\left(t\right)=\overset{\alpha \neq \mu }{\underset{\alpha }{\sum }}\left[K_{\mu \alpha }G_{\alpha \nu }\left(t\right)-K_{\alpha \mu }G_{\mu \nu }\left(t\right)\right]$$
where $G_{\mu \nu }\left(t\right)$ is the probability for an exciton population to be in the *μ*th state at time *t* when initially in the *ν*th state at time t=0. $K_{\mu \nu }$ is the integral kernel, which is calculated in the second order to the coupling $H_{ext-ph}$ between the excitons and the bath. The numerical integration of $K_{\mu \alpha }$ is the most time consuming procedure. It is required to have two steps of evaluation:(10)$$K_{\mu \nu }=\int_{0}^{\infty }d\tau \left[K_{\mu \nu }^{L}\left(\tau \right)+K_{\mu \nu }^{L}\left(-\tau \right)\right]$$
(11)$$K_{\mu \nu }^{L}\left(\tau \right)=K_{\mu \nu }^{F}\left(\tau \right)[{\ddot{g}}_{\mu \nu \nu \mu }\left(\tau \right)-\left({\dot{g}}_{\nu \mu \nu \nu }\left(\tau \right)-{\dot{g}}_{\nu \mu \mu \mu }\left(\tau \right)+2i\lambda_{\nu \mu \nu \nu }\right)\;\left({\dot{g}}_{\nu \nu \mu \nu }\left(\tau \right)-{\dot{g}}_{\mu \mu \mu \nu }\left(\tau \right)+2i\lambda_{\mu \nu \nu \nu }\right)]$$
(12)$$K_{\mu \nu }^{F}\left(\tau \right)=exp\left[-i\left(\epsilon_{\mu }-\epsilon_{\nu }\right)\tau -g_{\mu \mu \mu \mu }\left(\tau \right)-g_{\nu \nu \nu \nu }\left(\tau \right)+g_{\nu \nu \mu \mu }\left(\tau \right)\;-g_{\mu \mu \nu \nu }\left(\tau \right)-2i(\lambda_{\nu \nu \nu \nu }-\lambda_{\mu \mu \nu \nu })\tau \right]$$
where $\epsilon_{\mu }$ and $\epsilon_{\nu }$ are the exciton states; $\ddot{g}\left(t\right)$ and $\dot{g}\left(t\right)$ are the second and the first derivatives; $\lambda_{\mu \nu \alpha \beta }=-\underset{\tau \to \infty }{\lim}Im\left[\frac{dg_{\mu \nu \alpha \beta }\left(\tau \right)}{d\tau }\right]$ is the reorganization energy in exciton representation [49].

Finally, the absorption, circular dichroism (CD), and fluorescence spectra of the PSI complex without the inhomogeneous broadening are written as follows:(13)$$\sigma_{abs}\left(\omega \right)\approx \frac{\omega }{\pi }\overset{N}{\underset{\alpha }{\sum }}d_{\alpha }^{2}Re\overset{\infty }{\underset{0}{\int }}dte^{i\left(\omega -\epsilon_{\alpha }\right)t}e^{-g_{\alpha \alpha \alpha \alpha }\left(t\right)}e^{-0.5K_{\alpha \alpha }t}$$
(14)$$\sigma_{CD}\left(\omega \right)\approx \frac{\omega }{\pi }\overset{N}{\underset{\alpha }{\sum }}R_{\alpha }Re\overset{\infty }{\underset{0}{\int }}dte^{i\left(\omega -\epsilon_{\alpha }\right)t}e^{-g_{\alpha \alpha \alpha \alpha }\left(t\right)}e^{-0.5K_{\alpha \alpha }t}$$
(15)$$\sigma_{fl}\left(\omega \right)\approx \frac{\omega^{3}}{\pi }\overset{N}{\underset{\alpha }{\sum }}\frac{{\left(nd_{\alpha }\right)}^{2}e^{\epsilon_{\alpha }\beta }}{\sum_{n}e^{\epsilon_{\alpha }\beta }}Re\overset{\infty }{\underset{0}{\int }}dte^{i\left(\omega -\epsilon_{\alpha }+2\lambda_{\alpha \alpha \alpha \alpha }\right)t}e^{-g_{\alpha \alpha \alpha \alpha }{}^{*}\left(t\right)}e^{-0.5K_{\alpha \alpha }t}$$
where **n** is a polarization unit vector, and $K_{\alpha \alpha }=\sum_{\beta }K_{\alpha \beta }$ are the exciton relaxation rates. $R_{\alpha }=\sum_{nm}c_{n}^{\alpha }c_{m}^{\alpha }r_{nm}\left(d_{n}\times d_{m}\right)$ is the rotational strength in the exciton representation, where $r_{nm}$ is the distance between the centers of transition moments.

To take the inhomogeneous broadening into account, we will use the Monte Carlo method. In the exciton Hamiltonian (6), for each $\Omega_{n}$ transition energy we consider the stochastic Gaussian distribution (${\mathrm{SD}}_{\Omega }={\mathrm{FWHM}}_{\Omega }/2\sqrt{2\ln2}$ the standard deviation) of the transition energy modulation Δ caused by the interaction with the environment. Applying a random number generator, the exciton Hamiltonian is diagonalized with a set of transition energies $\Omega_{n}+\Delta_{n}$ for each Monte Carlo realization. The final spectra are calculated by averaging over all Monte Carlo realizations:(16)$$\sigma_{abs}\left(\omega \right)\approx \langle \frac{\omega }{\pi }\overset{N}{\underset{\alpha }{\sum }}d_{\alpha }^{2}Re\overset{\infty }{\underset{0}{\int }}dte^{i\left(\omega -\epsilon_{\alpha }\right)t}e^{-g_{\alpha \alpha \alpha \alpha }\left(t\right)}e^{-0.5K_{\alpha \alpha }t}\rangle {}_{\Delta }$$
(17)$$\sigma_{CD}\left(\omega \right)\approx \langle \frac{\omega }{\pi }\overset{N}{\underset{\alpha }{\sum }}R_{\alpha }Re\overset{\infty }{\underset{0}{\int }}dte^{i\left(\omega -\epsilon_{\alpha }\right)t}e^{-g_{\alpha \alpha \alpha \alpha }\left(t\right)}e^{-0.5K_{\alpha \alpha }t}\rangle {}_{\Delta }$$
(18)$$\sigma_{fl}\left(\omega \right)\approx \langle \frac{\omega^{3}}{\pi }\overset{N}{\underset{\alpha }{\sum }}\frac{{\left(nd_{\alpha }\right)}^{2}e^{\epsilon_{\alpha }\beta }}{\sum_{n}e^{\epsilon_{\alpha }\beta }}Re\overset{\infty }{\underset{0}{\int }}dte^{i\left(\omega -\epsilon_{\alpha }+2\lambda_{\alpha \alpha \alpha \alpha }\right)t}e^{-g_{\alpha \alpha \alpha \alpha }{}^{*}\left(t\right)}e^{-0.5K_{\alpha \alpha }t}\rangle {}_{\Delta }$$


### 7.3. Computational Parameters for PSI Monomer and Trimer Models

The line-shape function of Chl, g(t), which is necessary to model the linear response (16) and the steady-state fluorescence (17), was already discussed in the previous chapter. Let us turn to the evaluation of the PSI exciton Hamiltonian (6). Two sets of parameters, the Qy Chl site energies $\Omega_{n}$, and the Chl-Chl interaction energies $J_{mn}$, have to be determined. The different ways of $J_{mn}$ estimation have been debated before [17,19,40]. It has been established that the dipole-dipole approximation is not good at all for the nearby pigments, particularly for the special pair and some dimeric and trimeric Chls in PSI. The extended dipole approximation proved to the best choice in terms of computational costs and accuracy [19,40]. Thus, we used this method in our study too.

The choice of site energies is the most critical procedure of the PSI optical response modeling. Based on the comparison of the existing crystal structures (*Thermosynechococcus* and *Synechocystis*) and on analysis of the microphotographs of the PSI trimers from *Arthrospira platensis*, we decided to consider 96 Chls in the PSI monomer from *Arthrospira platensis*, the same as for that from *Thermosynechococcus*. Surely, the real number of pigment molecules can be established after crystallization of the complex, but for now we will use this number. The set of $\Omega_{n}$ is given in Table A2. The site energies for bulk Chls, which contribute to the main PSI absorption band, are in the range from 14920 to 15080 cm^−1^. Taking into account the fact that the inhomogeneous broadening is about 200 cm^−1^, it is not possible to determine the site energies more precisely than ±50 cm^−1^ by fitting the experimental data.

There are several hypotheses concerning the assignment of red Chls in the cyanobacterial PSI. However, the situation could be more complicated: different groups of Chls may be responsible for the long-wavelength absorption in different bacteria. For example (Figure 2e), some differences between PSI monomer crystal structures of *Thermosynechococcus* and *Synechocystis* are exactly in the B31-B32-B33 trimer (a lack of one Chl), which is considered as one of the putative candidates for red Chls in *Thermosynechococcus*. The SMS studies of PSI trimers from *Arthrospira platensis* with oxidized P700 at 1.4 K [14] revealed two distinctive bands at 726 nm and 760 nm and an additional one at 714 nm. It was shown that the zero phonon line (ZPL) occurs only in the spectra in the range of 698–740 nm. The occurrence of the ZPL is directly connected with exciton-phonon coupling. If the coupling is strong, the phonon wing is dominant in the spectrum, but if the coupling is weak, the Huang-Rhys factor S<0.7, and ZPL is visible. The Huang-Rhys factor for the red states with weak coupling has been deduced [14] as S=0.35±0.05. Thus, in our modeling we will reproduce the ZPL distribution only for the exciton states with weak exciton-phonon coupling.

In selecting pigment molecules for the role of red Chls, we were guided by two considerations. (i) Analysis of the CD and LD spectra gives evidence for heterogeneity of the site energies of Chls [19]. It was concluded that the absorption of red Chls is already red-shifted without exciton coupling. (ii) Additional detergent treatment of the PSI monomeric and trimeric complexes allowed us to delete a certain amount of the peripheral Chls and, perhaps, some small peripheral subunits. It turned out that the long-wavelength bands not only remained in the absorption spectra of PSI after the treatment, but their intensities increased in comparison with the main peak at 680 nm (unpublished data). We have assumed that this effect is due to the fact that the putative red Chls are located in the proximity of the reaction center and cannot be easily removed by the detergent treatment. Thus, we propose an exciton model of the PSI monomer from *Arthrospira platensis* in which the red Chls are A40, B39, B26, and A28, and for the PSI trimer the additional states L01 and L02 form the PsaL subunit (Figure 9).

## 8. Conclusions

To interpret the single molecule fluorescence spectroscopy data obtained for the PSI trimeric complexes isolated form *Arthrospira platensis*, an exciton model has been developed based on modified Redfield theory of energy transfer in molecular aggregates. This model includes the effects of inhomogeneous broadening and the electron-phonon coupling that is introduced by the spectral density function. A total of 49 phonon modes were used to calculate the chlorophyll temperature-dependent line-shape function. This function was applied to model the exciton states absorption and fluorescence spectra.

The set of Qy site energies for the PSI monomer was determined by simultaneous fitting of the linear and the single molecule spectroscopy data. This set of site energies includes the energies of red Chls, which are supposed to be in the close proximity of the reaction center. The PSI trimer site energies require additional modifications of the PsaL subunit Chls. It turned out that the interaction energy between PSI antenna pigments cannot be considered as a key factor for explanation of the red Chls phenomenon. The exciton spectra were calculated, taking into account weak electron-phonon coupling, while the broad spectra of the F760 emitter were calculated considering strong electron-phonon coupling to reproduce the rather large Stokes shift.

Taking into account our findings, it is clear that additional modeling of the PSI trimer transient absorption experiments with closed and open RC is necessary to properly reproduce the SMS fluorescence lineshape and identify the pigments that correspond to the far red F726 and F760 emitters.

## Figures and Tables

**Figure 1 molecules-24-00822-f001:**
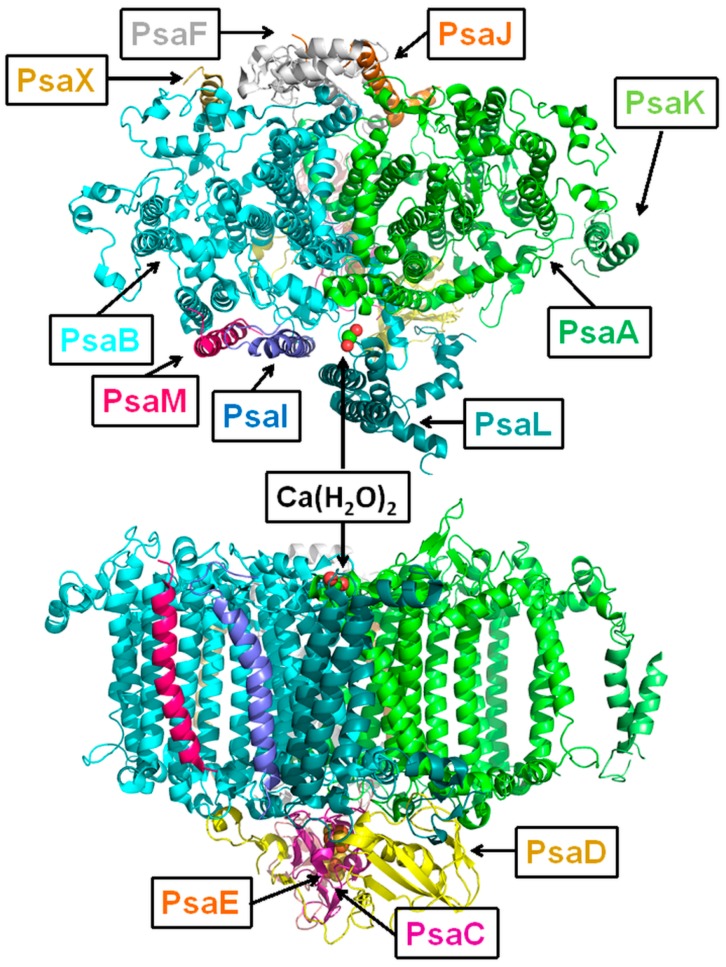
The transmembrane helices of the PSI monomer complex from *Thermosynechococcus* are shown without cofactors from the stromal side (upper picture). The side view is in the lower picture. The names of protein subunits are given in text boxes. The color of the text corresponds to the color of subunit.

**Figure 2 molecules-24-00822-f002:**
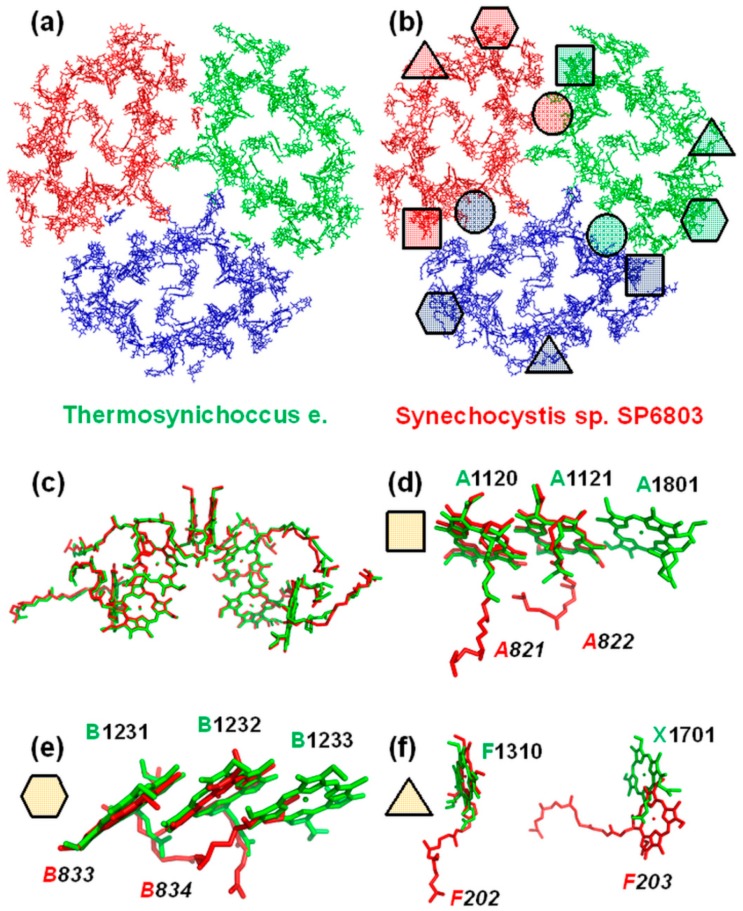
Differences of Chl arrangement in the PSI trimers isolated from *Thermosynechococcus e.* (**a**) and *Synechocystis sp. SP6803* (**b**). The data were taken from the published crystal structures: 1JB0.pdb and 5OY0.pdb. The locations of RC cofactors are almost identical (**c**). The missing Chls in the Synechocystis structure are marked with geometric figures (**d**–**f**).

**Figure 3 molecules-24-00822-f003:**
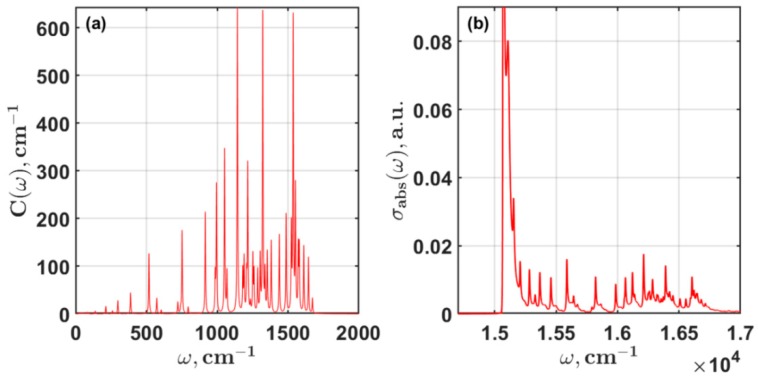
Chl a spectral density (**a**) and zero phonon line (**b**) at 4 K calculated according to Equation (3).

**Figure 4 molecules-24-00822-f004:**
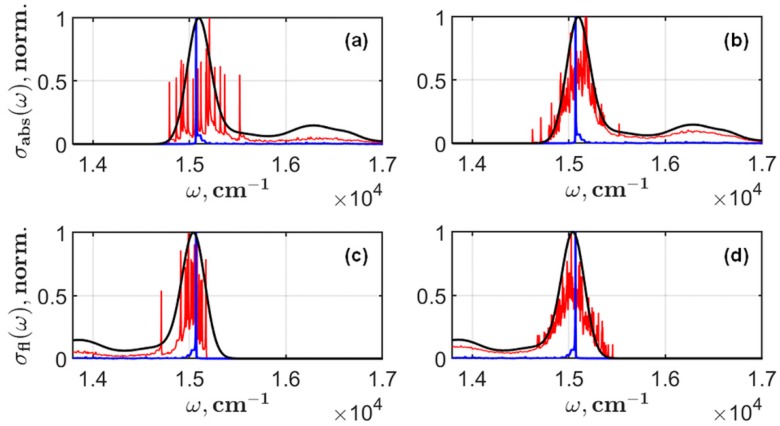
Calculated absorption (**a**,**b**) and steady-state fluorescence (**c**,**d**) of monomeric Chl in dimethyl ether are presented. The zero phonon line (blue) and the calculated (black) spectra according Equations (1) and (2) are shown for comparison with the spectra (red) obtained by using the Monte Carlo method for simulation of the inhomogeneous broadening effect. These spectra (red) were obtained after 20 (**a**,**c**) and 200 (**b**,**d**) Monte Carlo runs.

**Figure 5 molecules-24-00822-f005:**
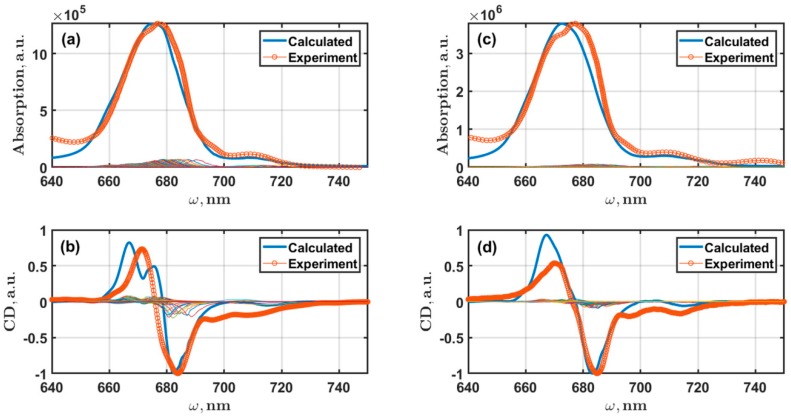
Calculated exciton absorption (upper plots) and CD (lower plots) spectra for PSI monomers (**a**,**b**) and trimers (**c**,**d**). The experimental absorption spectra measured at 4 K; CD spectra measured at 77 K. The absorption and CD spectra were modeled for the corresponding temperatures. FWHM of inhomogeneous broadening is 200 cm^−1^. All data averaged over 1500 realizations of the Monte Carlo method. The thin colored lines are the intensities of exciton states.

**Figure 6 molecules-24-00822-f006:**
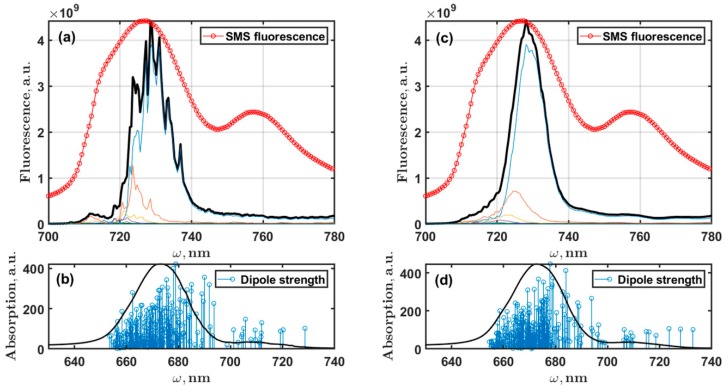
PSI trimer exciton fluorescence spectra calculated for 50 (**a**) and 1500 (**c**) realization of the Monte Carlo method are presented (black). The averaged SMS fluorescence experimental spectrum (red) is shown for comparison. The thin colored lines are the intensities of the exciton states. In addition to fluorescence spectra, the $\epsilon_{\alpha }-\lambda_{\alpha \alpha \alpha \alpha }$ vales (stick spectrum) for a particular realization of the inhomogeneous broadening are marked in blue (**b**, **d** lower plots) together with the absorption lineshape on the background. The length of the sticks is proportional to the dipole strength of the exciton states.

**Figure 7 molecules-24-00822-f007:**
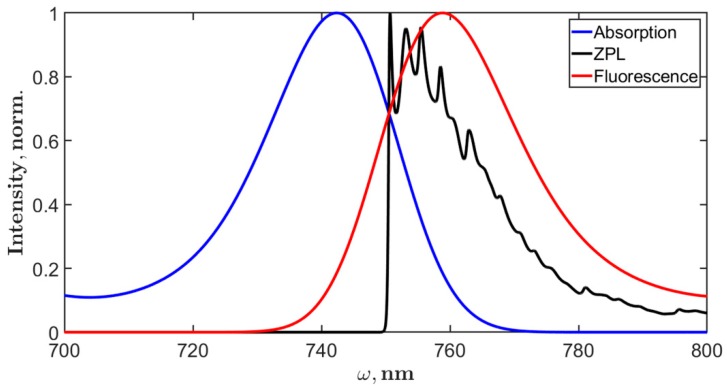
Calculated zero phonon line (ZPL), absorption, and fluorescence of F760 emitter. The shape of ZPL corresponds to the strong electron-phonon coupling. The inhomogeneous broadening is 200 cm^−1^. The resulting Stokes shift is about 20 nm.

**Figure 8 molecules-24-00822-f008:**
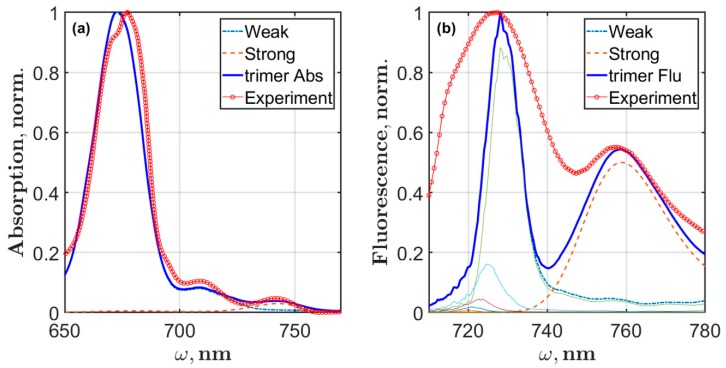
Results of Qy linear absorption (**a**, trimer Abs) and steady-state fluorescence (**b**, trimer Flu) modeling for the PSI trimer isolated from *Arthrospira platensis*. The excitonic part (Weak) of both spectra is calculated assuming weak electron-phonon coupling, whereas the F760 state contribution (Strong) requires the strong electron-phonon coupling approximation. The experimental cryogenic absorption and SMS fluorescence are shown (red). The thin colored lines are the intensities of exciton states.

**Figure 9 molecules-24-00822-f009:**
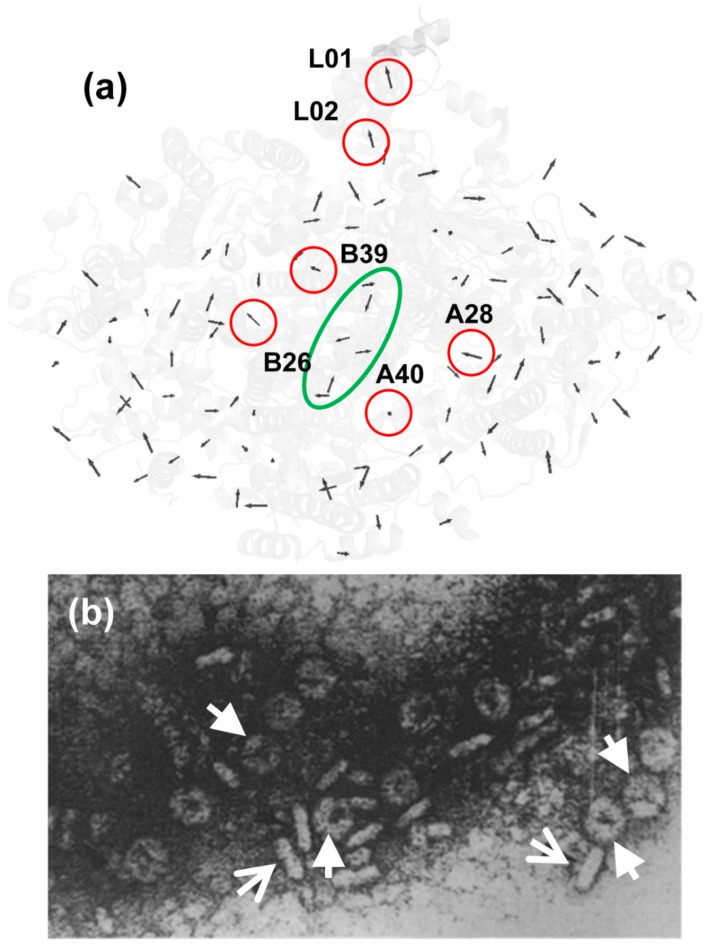
The Qy transition moments of Chl in the proposed monomer model for PSI isolated from *Arthrospira platensis* are shown (**a**). The Qy transitions of the assumed red Chls are marked with red circles. The green ellipse marks the reaction center. Micrograph of PSI trimers form *Arthrospira platensis* (**b**) taken from the literature [21]. The arrows indicate the top and side views of the trimers.

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
