# Peer review of "Single Molecule Fluorescence Spectroscopy of PSI Trimers from Arthrospira platensis: A Computational Approach"

_molecules, 2019, doi:10.3390/molecules24040822_

Reviewer 1 Report

This manuscript is original and useful for plant scientists.. 

As whole the paper is logically designed to reach main goals of the research work.

The abstract clearly describes purpose of study and main findings. Title is clear and precise.

Introduction is a little descriptive. Attractive hypothesis is missing in this chhapter.

The discussion is linked to the main results and the purpose of the study. However, conclusion should be toned down and .

The presentation of the results and figures facilitated the  understanding of mechanisms.

The main comment to the text that in part of introduction is missing informative part about functions of PSI and connectivity with PSII under different environmental stress.

The paper is clear and well-written. Style is adequate. Length of the paper is appropriate.

The paper is interesting and approaches a very important issue, nevertheless I have serious concerns on the novelty. Authors could highlight strong but I would like to invite authors  to discuss more about functional aspects of

Zivcak M. et al. : Repetitive light pulse-induced photoinhibition of photosystem I severely affects CO2 assimilation and photoprotection in wheat leaves. Photosynthesis Resarch 2015, DOI 10.1007/s11120-015-0121-1

Huang W, et al: Cyclic electron flow plays an important role in photoprotection for the resurrection plant Paraboea rufescens under drought stress. Planta 2012, 235, 819-828.

Shikanai T.: Regulatory network of proton motive force: contribution of cyclic electron transport around photosystem I. Photosynthesis Research 2016, 129, 253–260

I recommend to ACCEPT the paper for publication, after MINOR REVISIONS related to some issues which are not fully clear and need to be explained in the discussion. 

Author Response

Comments and Suggestions for Authors

This manuscript is original and useful for plant scientists.. 

As whole the paper is logically designed to reach main goals of the research work.

The abstract clearly describes purpose of study and main findings. Title is clear and precise.

Introduction is a little descriptive. Attractive hypothesis is missing in this chapter.

Response 1. Checked. The corresponding changes concerning the existing publications in the field and attractive hypothesis are made in the Introduction:

Line 88:a detailed exciton model of the energy transfer and charge separation has to be developed. The fitting of linear and the steady-state fluorescence spectra of PSI form Thermosynechococcus elongatus based on the Redfield theory has already been done [11]. The site energies of Chls have been determined by using the genetic optimization algorithm. Some thorough calculations on PSI, when the site energies were calculated applying various modifications of semiclassical approximations, have also been performed [12,13,29-31]. In these studies several solutions for the two main difficulties of PSI theory making: the Chl Qy site energies identification, LWCs assessment and the Chl mutual interaction energy calculation, have been proposed. The main drawback of almost all these studies is a poor elaboration of the exciton theory which is necessary to model the realistic optical response. Without additional consideration and analysis of the electron-phonon coupling and the relaxation rates of the PSI exciton manifold, it will not be possible to link together the numerous experimental data and the theoretical estimates of Qy transition energies of Chls and the couplings between them.

The aim of this research is to propose a reasonable interpretation of the LWCs fluorescence dynamics at cryogenic temperatures of the PSI trimer isolated from Arthrospira platensis, taking into account the existing crystal structure for Thermosynechococcus and Synechocystis, as well as our preliminary studies on PSI exciton models [32]. Considering the outcomes of the SMS experimental work [8] on PSI form Arthrospira platensis, we implement in our theory the different regimes of electron-phonon coupling for the excitons and for LWCs. To achieve the desired goal, for the first time, we are going to apply the modified Redfield theory for the exciton model development and the multimode Brownian oscillator model for the Qy absorption of Chl modeling.

The discussion is linked to the main results and the purpose of the study. However, conclusion should be toned down and.

Response 2: Checked. The same remark was made by the third reviewer. The detailed response on this remark will be given in the reply to the comments of the 3rd reviewer.

The presentation of the results and figures facilitated the understanding of mechanisms.

The main comment to the text that in part of introduction is missing informative part about functions of PSI and connectivity with PSII under different environmental stress.

Response 2: Checked. The corresponding changes are made in the Introduction:

Line 45: … Along with the photosystem II (PSII), cytochrome b6f and the ATP synthase, PSI is an indispensable component of the photosynthetic apparatus. Converting the solar energy into the chemical bond energy, PSI and PSII act in tandem. The amount of PSI in cyanobacteria is relatively higher in comparison with PSII. The PSI/PSII ratio varies from about 3 up to 5.8 [3,4]. PSII absorbs a phono with the wavelength less than 680 nm creating the cation radical P680+ which drives the sequential oxidation of water. The PSI in turn captures the light energy up to 700 nm. The oxidizing potential for the primary electron donor of PSI is not enough to obtain electrons from water; however, it is enough to support the electron transfer between cytochrome and ferrodoxin. Thus along with the linear electron transport from water to NADP+, PSI supports the cyclic electron transport through membrane [5], which plays an important role in photoprotection [6,7].

The paper is clear and well-written. Style is adequate. Length of the paper is appropriate.

The paper is interesting and approaches a very important issue, nevertheless I have serious concerns on the novelty.

Response 3: Checked. The novelty of the paper is emphasized in the final part of the Introduction. See Response 1.

Authors could highlight strong but I would like to invite authors to discuss more about functional aspects of

Response 4: Checked. See the Response 2.

Zivcak M. et al. : Repetitive light pulse-induced photoinhibition of photosystem I severely affects CO2 assimilation and photoprotection in wheat leaves. Photosynthesis Resarch 2015, DOI 10.1007/s11120-015-0121-1

Response 5: The reference is included in the reference list.

Huang W, et al: Cyclic electron flow plays an important role in photoprotection for the resurrection plant Paraboea rufescens under drought stress. Planta 2012, 235, 819-828.

Response 6: The reference is included in the reference list.

Shikanai T.: Regulatory network of proton motive force: contribution of cyclic electron transport around photosystem I. Photosynthesis Research 2016, 129, 253–260

Response 7: The reference is included in the reference list.

 Reviewer 2 Report

In this manuscript a computational approach is used to calculate the CD, absorption and fluorescence spectra of Photosystem I from two cyanobacteria species. The work is of potential interest for the journal Molecules, but two main points should be addressed. One, based on the title and the text one would expect that the manuscript is about single molecule fluorescence spectroscopy but this data is not presented in the entire manuscript. Second, the English of the manuscript should be edited. A list of examples of mistakes in the English and other minor points are given below:

14. single molecular -> single molecule

32. which is entirely depends -> which entirely depends

33. The pigment-protein complexes (PPC)

35. PBC -> PPC

40. “LHCs include up to hundreds of pigments”, do you mean per RC or per protein?

50. Several bacterial species, or just two?

58. the energy trapping is -> the energy trapping does

71-73: How does sample preparation of SMS standout from the other techniques, and how is the experimental data interpretation different?

74. Better Signal to noise ratio compared to single molecule absorption?

102. Fe4S4 should be Fe4S4.

121. under the low light -> under low light

121-122 add reference

138. which are might -> which might

272. choce -> choice

294. is appeared -> is visible

307. terminal -> peripheral

311. washed out -> removed

333. Could you show these single molecule results and compare with your data?

 Author Response

Comments and Suggestions for Authors

In this manuscript a computational approach is used to calculate the CD, absorption and fluorescence spectra of Photosystem I from two cyanobacteria species. The work is of potential interest for the journal Molecules, but two main points should be addressed. One, based on the title and the text one would expect that the manuscript is about single molecule fluorescence spectroscopy but this data is not presented in the entire manuscript. Second, the English of the manuscript should be edited. A list of examples of mistakes in the English and other minor points are given below:

14. single molecular -> single molecule

Response 1: Checked. The text of the manuscript is modified.

32. which is entirely depends -> which entirely depends

Response 2: Checked. The text of the manuscript is modified.

33. The pigment-protein complexes (PPC)

Response 3: Checked. The text of the manuscript is modified

35. PBC -> PPC

Response 4: Checked. The text of the manuscript is modified

40. “LHCs include up to hundreds of pigments”, do you mean per RC or per protein?

Response 5: Checked. We meant “hundreds of pigments per protein”. Some LHCs have no RC, for instance, LH2 of purple bacteria.

50. Several bacterial species, or just two?

Response 6: Checked. Just two bacterial species.

58. the energy trapping is -> the energy trapping does

Response 7: Checked. The text of the manuscript is modified

71-73: How does sample preparation of SMS standout from the other techniques, and how is the experimental data interpretation different?

Response 8: Checked. The text of the manuscript is modified by adding the following:

A sample under study is usually in a buffer solution, however in the case of SMS, it has to be immobilized on a surface. The isolated PPCs are exhibited the fluorescence blinking which is influenced by the slight changes of protein conformation and variations of coupling with the local environment [].

74. Better Signal to noise ratio compared to single molecule absorption?

Response 9: Checked. In comparison with linear absorption. The text of the manuscript is modified

102. Fe4S4 should be Fe4S4.

Response 10: Checked. The text of the manuscript is modified

121. under the low light -> under low light

Response 11: Checked. The text of the manuscript is modified

121-122 add reference

Response 12: Checked

Grotjohann, I.; Fromme, P. Structure of cyanobacterial Photosystem I. Photosynthesis Research 2005, 85, 51-72,

138. which are might -> which might

Response 13: Checked. The text of the manuscript is modified

272. choce -> choice

Response 14: Checked. The text of the manuscript is modified

294. is appeared -> is visible

Response 15: Checked. The text of the manuscript is modified

307. terminal -> peripheral

Response 16: Checked. The text of the manuscript is modified

311. washed out -> removed

Response 17: Checked. The text of the manuscript is modified

333. Could you show these single molecule results and compare with your data?

Response 18: Checked. The third reviewer asked the same question. We took the data form ref. 8. (Brecht, M.; Hussels, M.; Schlodder, E.; Karapetyan, N.V. Red antenna states of Photosystem I trimers from Arthrospira platensis revealed by single-molecule spectroscopy. Biochimica Et Biophysica Acta-Bioenergetics 2012, 1817, 445-452). The fluorescence profile averaged over all PSI trimers (about 100) has been used for comparison with the calculated fluorescence spectra (inhomogeneous broadening is included in the model).

Reviewer 3 Report

In the presented manuscript Pishchalnikov et al. construct a Frenkel-exciton model of photosystem I. They calculate cryogenic-temperature linear absorption, circular dichroism and fluorescence spectra of PSI monomers and trimers, averaged over a static energetic disorder. In relation to existing single-molecule spectroscopy experiments, they discuss the effect of number of realizations in the averaging on the spectral shape. The authors also propose an assignment of the locations of the low-energy chlorophylls and calculate their emission spectrum.

First of all, the achievement of simulating a system as large as PSI (288 chlorophyllls in a trimer!) has to be appreciated. The level of the theory - Frenkel-exciton model, highly-structured spectral density, modified Redfield dynamics - is satisfactorily high and the results look very reasonable. The model provides interesting physical insight into the excitonic structure of PSI and as such is of interest to the readers. However, the authors need to significantly improve the comparison with experimental data, the formulation of the outcome of the work, and the discussion of the treatment of the low-energy states. These issues must be addressed before the paper can be published. 

Major issues

First, the authors must elaborate on the comparison with experimental data. Although the paper is entitled “The single molecule fluorescence spectroscopy of PSI”, there are no calculated quantities which can be directly compared to the single-PSI experiments. Take for instance the paper by Brecht et al., BBA 1817, 445 (2012), Ref. 8, cited as a motivation for the study. There, spectra for different PSI trimers, FL peak distributions, and also spectra with various integration times are presented. These should, in principle, correspond to the spectra in Fig. 7 of the reviewed manuscript, where the FL spectrum is presented, averaged over low and high number of realizations. Why don’t the authors present the quantities corresponding to the experimental ones? For linear spectra, the fit to experiment is crucial. The authors compare only the absorption spectrum (Figs. 6 and 9), where the fit is reasonable but not excellent. Meanwhile, both the FL and CD spectra are available in literature (already discussed Ref. 8 and other papers, also by the authors of the current manuscript). While the peak positions seem reasonable both for CD and FL, the spectral shape looks quite different. Even in modelling of complexes with a few pigments, there typically exist several sets of site energies which produce the same absorption and FL spectra. In a complex with such a high number of pigments as here, various sets of site energies will probably produce not only similar spectra, but also dynamics. The authors should comment on the sensitivity of their model on its parameters and its uniqueness. The reader needs to know that (or how well) the model works before conclusions based on it are made.

Second, the authors should clearly formulate the conclusions of their work. The theory level is quite high and, as a result, its information content is very rich. The authors, however, do not seem to make use of this richness and to discuss the interesting results of their model. Their “Conclusions” section (number 5, following number 9 for discussion) is a summary of the work done. The parameters of the model themselves are, however, not a sufficient conclusion, especially when their uniqueness and importance is questionable (see above). The authors should discuss the meaning of the system parameters: is there an energy funnel to the RC? What is the excitonic structure (localization length etc.)? Are the high-frequency vibrations important? What is the role of the position of the low-energy pigments? How heterogeneous is the ensemble of the PSI trimers in its excitonic properties?

Third, the treatment of the low-energy chlorophylls, giving rise to the F760 emission band, is somewhat phenomenological and should be justified. As the authors mention, these chlorophylls cannot be in thermal equilibrium with the higher-energy pigments. This has to be a consequence of a kinetic argument. In the modified Redfield approach, the authors calculate the rates between the excitons. Are these slow enough to be responsible for the nonequilibrium behavior? Do they correspond to the ratio of I_F760/Iext? If yes, it is a remarkable result worth mentioning. If not, it should be discussed why. What allows the authors to treat the low-energy chlorophylls separately, with different parameters? Furthermore, the assignment of the low-energy chlorophyll positions must be discussed in the context of existing studies. There are several such assignments present in the literature (for example Yin et al., J. Phys. Chem. B 111, 9923 (2007) (Quantum Chemistry), Hatazaki et al., J. Phys. Chem. Lett. 9, 6669 (2018) (SMS study)). How do these compare to the ones used in this work?

Minor issues

The treatment of the static disorder/inhomogeneous broadening:

For individual chl molecules, the authors treat the inhomogeneous broadening analytically, assuming a Gaussian broadening (Eqs. 1,2). For the excitonic states, the authors claim that they can also add this broadening (page 8, r. 262). Is this true? The static disorder changes the structure of the excitonic states (delocalization, transition dipoles…), not only their linewidths. Can one still include the inhomogeneous broadening analytically like this? In contrast, on page 10, r. 338, the authors say that they use the Monte Carlo procedure instead of the analytical expression. Further on (Fig. 7), the MC averaging is also used. Is it used instead of or in addition to the analytical broadening?

Terms in expressions:

Eq. 7: are also the m not equal to n terms included? In that case the electron-phonon coupling is off-diagonal, causing transitions between the electronic states. In a standard treatment, the H_ext-ph is diagonal in the site (diabatic) basis. Is this a typo, or do the authors really include such terms? 

Eqs. 13,14,15: for the absorption and CD spectra, is there a factor of omega missing (can be seen from Maxwell equations)? For all lineshape expressions, there should be a factor of ½ in front of the lifetime broadening exp(-Kt) (can be seen from writing the wavefunction as |psi>=a|0>+b|1>,  populations decay as |a|^2 and |b|^2, while coherences decay as ab*, and in the lineshape case there is only the excited state |1> relaxation). Could this perhaps be the reason why their spectra seem broader than the experiments? Would including the ½ and increasing the static disorder help?

The temperature dependence:

The authors say, on page 11, r. 351, that they cannot calculate at 1-3K because their integrals do not converge. Consequently, they do say they do not try to optimize the fits. However, in the very next sentence, they argue that the difference is minor and experiments are very similar. The temperature difference is thus not a sufficient excuse for the lack of fit. Regarding the convergence of the integrals: from the power-of-2 number of time points, I assume the authors used FFT to calculate the spectra. Did they consider calculating in a rotating frame (times exp(iw_0t) and then shifting)? This would produce less oscillatory terms, enabling larger time steps and consequently longer integration times. Also, what do the authors mean by the computational costs increase much faster than expected?

The language:

Overall the language of presentation is well understandable. There are, however, some systematic errors, typically with an extra auxiliary verb such as “are disappeared”, “is depends” etc. The first sentence of the paper should be rephrased not to discourage the reader. 

The figures:

Some of the figure legends are incomplete. Figures 6, 7, 9b: what are the colorful lines? Fig. 9a: “Data” means the same as the “Experiment” in Fig. 6?

 Author Response

For Major issues, please see the attachment.

Minor issues

The treatment of the static disorder/inhomogeneous broadening:

For individual chl molecules, the authors treat the inhomogeneous broadening analytically, assuming a Gaussian broadening (Eqs. 1,2). For the excitonic states, the authors claim that they can also add this broadening (page 8, r. 262). Is this true? The static disorder changes the structure of the excitonic states (delocalization, transition dipoles…), not only their linewidths. Can one still include the inhomogeneous broadening analytically like this? In contrast, on page 10, r. 338, the authors say that they use the Monte Carlo procedure instead of the analytical expression. Further on (Fig. 7), the MC averaging is also used. Is it used instead of or in addition to the analytical broadening?

Respond: (equations are not in the proper form, see the attached file for normal view)

A very good comment. Thanks. The calculation of inhomogeneous broadening for the exciton states has been discussed very poorly and must be rewritten. The Monte Carlo procedure has been used t in the following way:

(i) in the exciton Hamiltonian (6), for each Qy transition energy, Ω_n, we considered the stochastic Gaussian distribution of the transition energy modulation  with the standard deviation SD_Ω=FWHM_Ω⁄(2√2ln2). Using the random number generator, the exciton Hamiltonian is diagonalized with the transition energies Ω_n+Δ_n for the each realization of MC.

(ii) the final spectra are calculated by averaging over all MC realizations:

σ_abs (ω)≈ω/π ∑_α^N▒d_α^2 Re∫_0^∞▒dte^i(ω-ϵ_α )t e^(-g_αααα (t) ) e^(-K_αα t) 〗〗〉_∆

σ_CD (ω)≈ω/π ∑_α^N▒R_α Re∫_0^∞▒dte^i(ω-ϵ_α )t e^(-g_αααα (t) ) e^(-K_αα t) 〗〗〉_∆

σ_fl (ω)≈ω^3/π ∑_α^N▒((nd_α )^2 e^(ϵ_α β))/(∑_n▒e^(ϵ_α β) ) Re∫_0^∞▒dte^i(ω-ϵ_α+2λ_αααα )t e^(-g_αααα^* (t) ) e^(-K_αα t) 〗〗〉_∆

This is the way one can include the static disorder effect (delocalization, transition dipoles and relaxation rates variations) on the exciton states. The corresponding changes are made in the text of the manuscript.

Terms in expressions:

Eq. 7: are also the m not equal to n terms included? In that case the electron-phonon coupling is off-diagonal, causing transitions between the electronic states. In a standard treatment, the H_ext-ph is diagonal in the site (diabatic) basis. Is this a typo, or do the authors really include such terms?

Respond: The off-diagonal exciton-phonon coupling is considered as a perturbation, while the diagonal part is treated non-perturbatively. The off-diagonal terms depend on the exciton wave function overlapping and the intensity of electron-phonon coupling. Such conditions allow us to get the realistic exciton rates in comparison with the Foerster and Redfield regimes. The details of this theory are described in the following publications:

Meier, T.; Chernyak, V.; Mukamel, S. Femtosecond photon echoes in molecular aggregates. Journal of Chemical Physics 1997, 107, 8759-8780.

Yang, M.N.; Fleming, G.R. Influence of phonons on exciton transfer dynamics: comparison of the Redfield, Forster, and modified Redfield equations. Chemical Physics 2002, 275, 355-372.

Zhang, W.M.; Meier, T.; Chernyak, V.; Mukamel, S. Exciton-migration and three-pulse femtosecond optical spectroscopies of photosynthetic antenna complexes. Journal of Chemical Physics 1998, 108, 7763-7774.

The comments are added to the text of the manuscript.

Eqs. 13,14,15: for the absorption and CD spectra, is there a factor of omega missing (can be seen from Maxwell equations)? For all lineshape expressions, there should be a factor of ½ in front of the lifetime broadening exp(-Kt) (can be seen from writing the wavefunction as |psi>=a|0>+b|1>,  populations decay as |a|^2 and |b|^2, while coherences decay as ab*, and in the lineshape case there is only the excited state |1> relaxation). Could this perhaps be the reason why their spectra seem broader than the experiments? Would including the ½ and increasing the static disorder help?

Respond: Checked. The ω factor of absorption and CD lineshape is included in our calculation. The missing ω in Eqs. (13), (14). is an error. The corresponding changes are made in the text of the manuscript.

The temperature dependence:

The authors say, on page 11, r. 351, that they cannot calculate at 1-3K because their integrals do not converge. Consequently, they do say they do not try to optimize the fits. However, in the very next sentence, they argue that the difference is minor and experiments are very similar. The temperature difference is thus not a sufficient excuse for the lack of fit. Regarding the convergence of the integrals: from the power-of-2 number of time points, I assume the authors used FFT to calculate the spectra. Did they consider calculating in a rotating frame (times exp(iw_0t) and then shifting)? This would produce less oscillatory terms, enabling larger time steps and consequently longer integration times. Also, what do the authors mean by the computational costs increase much faster than expected?

Respond: Calculating the modified Redfield absorption, we do the FFT and it is the easiest and cheapest computational procedure. There is no problem with FFT transformation – it is not produced the spurious oscillation. The problems arise when g(t) function (Eq. 3) has to be calculated. The numerical integration of this equation is very sensitive to the number of points at time scale and to the integration step. It is exactly the function which gives the temperature dependence. It has to be also taken into account that the convergence of this integral depends on the spectral density. For example, in the case of strong electron-phonon interaction a “clean” absorption line shape appears even at “rough” integrational conditions.

About “the computational costs increase much faster than expected”

The PSI trimer contains about 300 exciton states and the lineshape has to be calculated for each exciton state. It means that if the time scale is changed, for instance, form 2^10 (1024) to 2^11 (2048), the integration arrays are changed for all exciton states (it is also required more memory to operate). But the most time consuming procedure is the exciton relaxation matrix evaluation (Eq 10). After numerous trials we have found that poor integration of this equation can screw up the final spectra dramatically.

The language:

Overall the language of presentation is well understandable. There are, however, some systematic errors, typically with an extra auxiliary verb such as “are disappeared”, “is depends” etc. The first sentence of the paper should be rephrased not to discourage the reader.

Respond: Taking into account the comments and suggestions of other reviewers, we have tried to fix as much as possible grammar errors.

Round  2

Reviewer 2 Report

One comment. Light-harvesting proteins do generally not coordinate hundreds of pigments per protein. This is why the statement about pigments per proteins should be rephrased into tens of pigments per protein.

Author Response

(i)            The manuscript is now acceptable for publications. Although it is still a good idea to check the grammar.

Response: the manuscript is sent for proofreading

(ii)          One comment. Light-harvesting proteins do generally not coordinate hundreds of pigments per protein. This is why the statement about pigments per proteins should be rephrased into tens of pigments per protein.

Response: Checked

Reviewer 3 Report

Based on the reports by the other two reviewers I expect the manuscript to be accepted. I can only join with my recommendation, the authors have addressed my remarks satisfyingly and the mansucript is now well-suited for publication in Molecules.